# Prognostic Significance of Peripheral Blood Parameters as Predictor of Neoadjuvant Chemotherapy Response in Breast Cancer

**DOI:** 10.3390/ijms26062541

**Published:** 2025-03-12

**Authors:** Ionut Flaviu Faur, Amadeus Dobrescu, Ioana Adelina Clim, Paul Pasca, Cosmin Burta, Cristi Tarta, Dan Brebu, Andreea-Adriana Neamtu, Vlad Braicu, Ciprian Duta, Bogdan Totolici

**Affiliations:** 1IInd Surgery Clinic, Timisoara Emergency County Hospital, 300723 Timisoara, Romania; flaviu.faur@umft.ro (I.F.F.); dobrescu.amadeus@umft.ro (A.D.);paul.pasca@umft.ro (P.P.); tarta.cristi@umft.ro (C.T.); brebu.dan@umft.ro (D.B.); braicu.vlad@umft.ro (V.B.); duta.ciprian@umft.ro (C.D.); 2X Department of General Surgery, “Victor Babes” University of Medicine and Pharmacy Timisoara, Eftimie Murgu Square 2, 300041 Timisoara, Romania; 3Multidisciplinary Doctoral School “Vasile Goldiș”, Western University of Arad, 310025 Arad, Romania; 4Doctoral School of Medicine, “Victor Babes” University of Medicine and Pharmacy Timisoara, Eftimie Murgu Square 2, 300041 Timisoara, Romania; 5Faculty of Medicine, “Victor Babes” University of Medicine and Pharmacy Timisoara, Eftimie Murgu Square 2, 300041 Timisoara, Romania; mihai.burta@student.umft.ro; 6Faculty of Pharmacy, “Victor Babes” University of Medicine and Pharmacy Timisoara, Eftimie Murgu Square 2, 300041 Timisoara, Romania; andreea.neamtu@umft.ro; 7Ist Clinic of General Surgery, Arad County Emergency Clinical Hospital, 310158 Arad, Romania; totolici.bogdan@uvvg.ro; 8Department of General Surgery, Faculty of Medicine, “Vasile Goldiș” Western University of Arad, 310025 Arad, Romania

**Keywords:** pathological complete response (pCR), neutrophil-to-lymphocyte ratio (NLR), platelet-to-lymphocyte ratio (PLR), systemic immune-inflammatory index (SII)

## Abstract

The standard treatment for breast cancer typically includes surgery, often followed by systemic therapy and individualized treatment regimens. However, there is growing interest in identifying pre-therapeutic biomarkers that can predict tumor response to neoadjuvant chemotherapy (NACT). This study systematically evaluated various analytical parameters, including age, TNM stage, histological type, molecular subtype, and several biomarker ratios, such as the platelet-to-lymphocyte ratio (PLR), neutrophil-to-lymphocyte ratio (NLR), lymphocyte-to-monocyte ratio (LMR), systemic immune-inflammatory index (SII), and prognostic nutritional index (PNI). We aimed to assess the predictive value of these parameters regarding the tumor’s response rate to NACT. The analysis revealed a statistically significant association between the pathological complete response—pCR (absence of any detectable cancer cells in the tissue following neoadjuvant chemotherapy (NACT))—rate and NLR in the subgroup with values between 1 and 3 (*p* = 0.001). The optimal cut-off for PLR was determined to be 120.45, with 80.55% of patients achieving pCR showing PLR values below this threshold (*p* = 0.000). Similarly, the LMR cut-off was found to be 12.34, with 77.77% of patients with pCR having LMR values below this threshold (*p* = 0.002). Additionally, lower pre-therapeutic values of NLR (*p* < 0.001), PLR (*p* = 0.002), SII (*p* = 0.001), and LMR (*p* = 0.001) were significantly correlated with pCR compared to the non-pCR subgroup (*p* < 0.005). These findings highlight the predictive potential of these biomarkers for achieving pCR following NACT. Our study supports the hypothesis that pre-therapeutic values of NLR, PLR, SII, and LMR can serve as predictive biomarkers for pCR in breast cancer patients undergoing NACT. However, the PNI did not demonstrate predictive potential in relation to pCR. These biomarkers may provide valuable insights into patient prognosis and guide personalized treatment strategies.

## 1. Introduction

Breast cancer is a malignant tumor that represents the leading cause of morbidity and mortality among women globally. Its increasing prevalence has designated it as a significant public health concern. Advances in precision medicine have led to improvements in treatment strategies for breast cancer [1]. Currently, the standard approach for treating breast cancer primarily consists of surgery, often accompanied by systemic therapy and tailored comprehensive treatment plans [2,3,4,5]. In cases involving locally advanced tumors, characterized by factors such as a tumor diameter greater than 5 cm, axillary lymph node metastasis, or unfavorable molecular types (like HER-2 positive or triple-negative), or in patients with a tumor size-to-breast volume ratio that complicates breast preservation, preoperative neoadjuvant chemotherapy is frequently preferred. This strategy aims to downstage the tumor and lower the recurrence rate, ultimately enhancing patient survival [6,7,8,9,10]. Prior research has established that overall survival (OS) and recurrence-free survival (RFS) in patients undergoing neoadjuvant chemotherapy (NACT) are closely linked to the treatment’s effectiveness [11]. Achieving pathological complete response (pCR) with NACT is typically associated with longer survival, making the early prediction of treatment efficacy in breast cancer crucial for personalized therapy [12].

Recently, the immune system has been recognized as a crucial factor in how breast cancer responds to chemotherapy [13,14]. Although breast cancer typically produces fewer neoantigens [15], it often exhibits significant infiltration by lymphocytes, known as tumor-infiltrating lymphocytes (TILs), which can vary considerably among different molecular subtypes [16,17]. The presence of TILs in breast cancer is strongly associated with achieving a pCR following neoadjuvant chemotherapy (NACT) [18,19,20]. Certain subtypes of tumor-infiltrating cells, such as regulatory T lymphocytes (Treg) and myeloid-derived suppressor cells (MDSCs), can contribute to immune suppression and have been shown to diminish the effectiveness of NACT in breast cancer cases [21].

Reflecting the immune system’s significant role, breast cancer has demonstrated a tendency to respond to immune checkpoint inhibitors (ICIs), although the effectiveness is generally lower compared to other cancers like melanoma, kidney, and lung cancer. Therefore, acknowledging the tumor immune microenvironment’s clear impact on breast cancer treatment responses, multiple research efforts are underway to elucidate the peripheral immune system’s role in breast cancer outcomes, particularly regarding responses to NACT [22].

Researchers have extensively studied various peripheral markers of immunity and inflammation, such as the neutrophil-to-lymphocyte ratio (NLR), platelet-to-lymphocyte ratio (PLR), platelet mean volume, and white blood cells-to-lymphocyte ratio, to explore potential correlations with pCR. Overall, lower values of these ratios suggest a systemic environment characterized by decreased inflammation and immune activation, which is associated with improved therapeutic responses [23]. However, the impact of these immunity and inflammation indicators on NACT responses has not yet been examined alongside other predictive factors, including molecular subtypes, tumor grading, and Ki67 levels.

## 2. Results

We analyzed a total of 142 patients who met the inclusion/exclusion criteria of the study. Thirty-six patients showed a complete pathological response (pCR) to neoadjuvant therapy (NACT), representing 25.35% of the total cohort. A total of 74.64% of the patients were classified as having a partial pathological response or stable disease, or as having progressive disease under neoadjuvant therapy. A statistically significant difference was observed between the complete pathological response rates and the incomplete pathological response rates (non-pCRs) in correlation with body mass index (BMI), such that patients with a BMI ≤ 30 kg/m^2^ showed better results in terms of complete pathological response to NACT (*p* = 0.005).

A significant pCR response rate was observed in patients initially staged as cT2 (*p* = 0.004), as well as in patients with tumor grading classified as G1 (*p* = 0.005) and G2 (*p* = 0.004), and also in patients diagnosed with ductal breast carcinoma (*p* = 0.005) (Table 1).

In the case of patients with pCR, based on the optimal cut-off values of the ROC curve, we established value subgroups for NLR, PLR, SII, and LMR. Regarding NLR, values below 1 were recorded in three cases (8.33%), values between 1 and 3 in 32 cases (88.8%), and values above 3 in one case (2.77%). pCR showed statistically significant data in relation to NLR in the subgroup with values between 1 and 3 (*p* = 0.001). The optimal cut-off value on the ROC curve for PLR was 120.45, so 29 cases (80.55%) out of the 36 with pCR had a PLR < 120.45, showing a statistically significant correlation (*p* = 0.000). Regarding SII, the cut-off value was found to be 403.4, with 31 patients (86.11%) out of the 36 with a pCR showing SII values below 403.4, which showed a statistically significant correlation for this parameter with the response rate to NACT therapy. The cut-off value for LMR was established according to the ROC curve as 12.34, so 28 patients (77.77%) out of the 36 with pCR had LMR values below 12.34 (*p* = 0.002) (Table 1).

Referring to the optimal cut-off values of the ROC curve, we divided the study group into two analysis subgroups: pre-therapeutic SII < 403.4 and >403.4. Statistically significant differences were obtained concerning the analytical parameters represented by age (*p* = 0.001), as well as the Luminal A molecular subtype (*p* = 0.001) and Luminal B (*p* = 0.004). Patients categorized as low SII predominantly had an age under 50 years and the Luminal A molecular subtype, while patients categorized as high SII were mainly over 50 years old, with the Luminal B molecular subtype. For other analysis parameters such as cTNM, Ki-67, and the molecular subtypes HER2++ and TNBC, no statistically significant data were obtained in correlation with the SII cut-off value (Table 2).

The analysis of the 142 cases that received NACT therapy allowed the stratification of the cohort into two study subgroups: non-pCR (106 cases, 74.64%) and pCR (36 cases, 25.35%). Dichotomous variables were analyzed using the χ^2^ test, while continuous variables were assessed with the t-test. We conducted both pre-therapeutic and post-therapeutic analyses of NLR, PLR, SII, PNI, and LMR. The analysis had a comparative nature between the non-pCR and pCR subgroups. Referring to the data obtained in Table 3, we can highlight that pre-therapeutic values of NLR (*p* < 0.001), PLR (*p* = 0.002), SII (*p* = 0.001), and LMR (*p* = 0.001) were lower in the pCR subgroup, obtaining a statistically significant coefficient in comparison to the non-pCR subgroup (*p* < 0.005). The comparative analysis of pre-therapeutic PNI and post-therapeutic PNI did not yield statistically significant data (*p* > 0.05) (Table 3).

Following the analysis of PLR, we identified 87 cases classified as PLR high (61.26%) and 55 cases classified as PLR low (38.73%). Of the 87 PLR high cases, 29 presented pCR (33.33%), while only 7 of the PLR low cases (12.72%) presented pCR (*p* = 0.006). We performed a univariate analysis of the NLR/PLR ratio, revealing the following: 39 cases (27.46%) were classified as NLR high/PLR high, 61 cases (42.95%) as NLR low/PLR high, 9 cases (6.33%) as NLR high/PLR low, and 33 cases (23.23%) as NLR low/PLR low. The NLR analysis highlighted that 49 cases were classified as NLR high (34.5%), 17 of which presented pCR (34.69%), while 93 cases were classified as NLR low (65.5%), with 19 presenting pCR (29.23%). A total of 78 cases (54.92%) were classified as LMR high, with 21 presenting pCR (26.92%), and 64 cases (45.08%) had LMR low, with 15 presenting pCR (23.43%).

The combined NLR and PLR analysis was significant for predicting pCR. Consistently, patients in the NLR high/PLR high subgroup had the highest pCR rate (52.7%), while those in the NLR low/PLR low subgroup had the lowest (5.55%). A high NLR (*p* = 0.004) and high PLR (*p* = 0.006) were associated with a higher pCR ratio compared to those with low NLR and low PLR. Similarly, a high LMR was associated with a higher pCR ratio compared to a low LMR (*p* = 0.005). Regarding the type of surgical intervention, 45 cases underwent modified radical mastectomy (Madden–Auchincloss type, 31.69%), of which 3 cases presented pCR (6.66%, *p* = 0.698). A total of 72 cases (50.7%) underwent conservative surgeries (sectoral resections, quadrantectomies), of which 22 cases (30.55%) presented pCR (*p* = 0.004, OR 1.177, 95% CI 0.211–7.276). A total of 25 cases (17.6%) underwent oncoplastic surgery, of which 11 (44%) presented pCR (*p* = 0.001, OR 1.744, 95% CI 0.211–7.276).

We also conducted a univariate analysis of the cohort, highlighting that patients with Ki-67 below 14% had a higher probability of being classified as NLR high, whereas patients with Ki-67 above 14% were more likely to be classified as NLR low (*p* < 0.05) (Table 4).

The multivariate analysis of the cohort in relation to pCR revealed statistically significant data regarding the luminal vs. non-luminal subtypes, Ki-67, tumor grading (G), chemotherapy regimen used, as well as PLR high, and NLR high/PLR high, which remained significant (Table 5). The Ki-67 ≤ 14% subgroup showed a five-fold higher pCR rate compared to patients with Ki-67 > 14% (OR 16.167, 95% CI 1.156–7.341, *p* = 0.018). Regarding tumor grading, significant statistical data were obtained for patients with G1/G2 tumors, who presented higher pCR rates compared to G3 tumors (OR 0.551, 95% CI 0.004–23.508, *p* = 0.026). Patients with NLR high and PLR high had a more than three-fold higher pCR rate compared to those with NLR low and PLR low (OR 0.156, 95% CI 0.078–0.912, *p* = 0.007). In terms of molecular subtype, the multivariate analysis highlighted superior pCR response rates in luminal subtypes compared to non-luminal subtypes (TN/HER2+) (OR 1.870, 95% CI 0.078–0.912, *p* = 0.005).

## 3. Discussion

### 3.1. Immune Response and Chemotherapy Effectiveness

Recent studies have highlighted the significant role of the immune response in influencing the effectiveness of chemotherapy for various tumors, including breast cancer. In particular, inflammatory markers, such as the neutrophil-to-lymphocyte ratio (NLR), have emerged as potential prognostic indicators. NLR reflects systemic inflammation, which is closely associated with tumor growth and metastasis [24]. Despite its potential, research on NLR as a prognostic marker in breast cancer remains limited, with mixed findings. However, preoperative blood tests—routinely conducted in breast cancer patients—offer a convenient, cost-effective way to assess inflammatory responses and could help predict treatment outcomes. In addition to NLR, other inflammatory markers, such as C-reactive protein (CRP), have been shown to play a critical role in evaluating breast cancer prognosis. These markers can serve as independent predictors of response to neoadjuvant chemotherapy (NACT). The heterogeneity in findings across different studies suggests that while these markers hold promise, there is currently no established absolute cut-off value or universally accepted gold standard for predicting chemotherapy response [25].

### 3.2. Previous Studies on NLR and PLR

A study by Xiaoyan Jin et al. in 2022 examined 67 breast cancer patients who received NACT. They performed a pre-therapeutic analysis of PLR and NLR and established cut-off values based on the ROC curve. Their results indicated that PLR may serve as a potential biomarker for predicting the effectiveness of NACT in breast cancer [26]. Specifically, logistic regression showed that high PLR values were correlated with superior pathological responses to NACT compared to low PLR values. Similarly, a 2022 study by Gaohua Yang et al. analyzed 95 breast cancer patients at the Hospital of Fujian Medical University. They assessed NLR, PLR, systemic immune-inflammatory index (SII), and prognostic nutritional index (PNI) in relation to NACT effectiveness, concluding that pre-therapeutic values of NLR, PLR, and SII could predict NACT response [27]. A larger retrospective study by Vincenzo Graziano et al., which included 373 breast cancer patients undergoing NACT, found no significant association between NLR or PLR and pathological complete response (pCR) (*p* > 0.05). However, a combined analysis of these markers showed that patients with both low NLR and low PLR had higher pCR rates than those with higher values of either marker (OR 2.29, 95% CI 1.22–4.27, *p* = 0.009) [28]. Our study mirrors this finding, demonstrating that the NLR low/PLR low profile was associated with superior pCR rates compared to higher values, reinforcing the predictive value of these markers.

### 3.3. Impact of Chemotherapy Regimen and Tumor Characteristics

The type of chemotherapy regimen may also influence the likelihood of achieving pCR, as more aggressive regimens could correlate with better pCR outcomes. However, further investigation is needed to understand how specific chemotherapy protocols impact the response. We observed that higher SII values were significantly correlated with Ki-67 > 14%, which suggests higher tumor proliferation. Luminal subtypes, including ER-positive and HER2-negative tumors, were more likely to achieve pCR compared to non-luminal subtypes like triple-negative breast cancer (TNBC) or HER2-positive tumors. This highlights the potential of luminal subtypes to respond more favorably to neoadjuvant chemotherapy. Additionally, our study found that patients with Ki-67 ≤ 14% had significantly higher pCR rates compared to those with Ki-67 > 14%. This suggests that lower proliferative activity, as indicated by Ki-67, may be a useful predictor of a favorable response to chemotherapy. Tumor grading also played a role in pCR outcomes. Patients with G1 and G2 tumors (low to moderate grade) presented higher pCR rates than those with G3 tumors (high grade). This finding supports the idea that tumors with lower grading are more sensitive to the effects of neoadjuvant chemotherapy.

### 3.4. Correlation Between Immune Markers and pCR

Our study confirmed that lower pre-therapeutic values of NLR, PLR, SII, and LMR were significantly associated with higher pCR rates. These immune-related biomarkers suggest that a favorable immune response may contribute to better chemotherapy outcomes. Furthermore, low SII values were associated with Ki-67 ≤ 14%, indicating that lower systemic inflammation might correlate with slower-growing tumors. This finding implies that SII could reflect tumor aggressiveness, with high SII values aligning with more proliferative and potentially more aggressive tumor behavior. High SII values were correlated with a lower likelihood of achieving pCR, possibly reflecting an immune profile that favors tumor growth or resistance to treatment. This supports the hypothesis that a favorable systemic immune environment is crucial for an optimal response to NACT. A meta-analysis by Xue Qi et al. in 2023, which included 22 studies with 5533 breast cancer patients receiving NACT, concluded that elevated PLR values were associated with a lower pCR rate (HR 0.77, 95% CI 0.67–0.88, *p* < 0.001) and worse overall survival (HR 1.97, 95% CI 1.56–2.5, *p* < 0.001) [29,30,31]. This highlights the importance of inflammatory markers in predicting chemotherapy outcomes and survival. Additionally, research by Coffelt et al. demonstrated that neutrophils contribute to tumor development and spread. The NLR, as a reflection of systemic immune responses influenced by cytokines, plays a critical role in tumor angiogenesis, survival, and proliferation [32]. Neutrophils and key signaling pathways, such as NF-κB and cytokines like TNF-α and IL-6, enhance tumor progression, which may explain the observed increase in NLR in more aggressive tumors. Platelets, which contribute to tumor growth and metastasis, also play a vital role in these processes [33].

## 4. Materials and Methods

### 4.1. Patient Selection

The current study includes 142 patients with histopathologically and immunohistochemically confirmed breast cancer diagnosis, who underwent neoadjuvant therapy (NACT) at the Pius Brinzeu County Emergency Clinical Hospital in Timisoara, Surgery Clinic II. The study lasted for a year and was conducted in accordance with the provisions of the 2013 Helsinki Declaration, respecting the ethical integrity of the patients and medical practice. The study was approved by the hospital’s ethics committee, according to approval No. 462/2024 dated 18 April 2024. Inclusion criteria for the study were as follows: (1) Cases confirmed histopathologically and immunohistochemically (core biopsy via Deltacut 14–16 G). (2) Cases that underwent molecular stratification (Luminal A (ER+, HER2−), Luminal B (ER+, HER2+, or HER2−); Luminal B tumors are also ER-positive, but they may either be HER2-positive or -negative, and they typically show higher Ki-67 expression compared to Luminal A tumors; HER2-positive subtype (HER2+), triple-negative breast cancer (TNBC); TNBC is characterized by the absence of ER, PR, and HER2 expression). (3) Age over 18 years and under 90 years. (4) Complete analysis of biological parameters (CBCs). (5) Cases that received neoadjuvant therapy (NACT). Exclusion criteria: (1) cases with incomplete oncological records; (2) age under 18 years and over 90 years; (3) refusal to participate in the study.

### 4.2. Clinical Characteristics

The collection of analytical parameters was carried out systematically and standardized, including age, TNM, histological type, molecular subtype, and biomarker ratios (platelet-to-lymphocyte ratio (PLR), neutrophil-to-lymphocyte ratio (NLR), and lymphocyte-to-monocyte ratio (LMR)), systemic immune-inflammatory index (SII), and prognostic nutritional index (PNI). Regarding the effectiveness and predictive nature of certain parameters, an exhaustive inter-relational analysis was performed in relation to the tumor response rate to NACT.

### 4.3. Methods

We conducted the study in Pius Brinzeu County Emergency Clinical Hospital in Timisoara, Surgery Clinic II. Peripheral venous blood samples (10 mL) and data collection peripheral venous blood samples were routinely obtained and measured within 1 week before NACT. The cells were analyzed by an XE-2100 hematology analyzer (Sysmex Corp., Kobe, Japan). All patient data included the clinical and pathological features, the type of treatment administered, and related outcomes.

### 4.4. Statistical Analysis

The database was processed using Microsoft Excel, and the data analysis was conducted with SPSS version 26.0 and Rapid Miner 9.6. Optimal cutoff values for the platelet-to-lymphocyte ratio (PLR), neutrophil-to-lymphocyte ratio (NLR), and lymphocyte-to-monocyte ratio (LMR) were determined through receiver operating characteristic (ROC) curve analysis. The area under the curve (AUC) was used to evaluate their predictive value, with the cut-off point selected as the ratio that provided the highest sensitivity and specificity. To assess the independent prognostic factors and the significance of PLR, NLR, systemic immune-inflammatory index (SII), and prognostic nutritional index (PNI) and LMR, both univariate and multivariate logistic regression models were applied. Odds ratios (ORs) and their 95% confidence intervals (95% CIs) were calculated. The relationships between NLR/PLR and pathological complete response (pCR), as well as other clinicopathological characteristics, were analyzed using Pearson’s χ^2^ test or Fisher’s exact test, as appropriate. A two-tailed *p*-value of less than 0.05 was considered statistically significant. Quantitative data with a normal distribution were expressed as mean ± standard deviation (x ± s). For group comparisons, a two-sample t-test was used, while the U test was applied to non-normally distributed data. Qualitative data were presented as counts and percentages, with comparisons between groups made using the chi-square test.

## 5. Conclusions

In our study, we highlighted that pre-therapeutic values of NLR, PLR, SII, and LMR were significantly correlated with pCR, confirming the predictive value and potential of these biomarkers. Regarding the analysis of PNI, we did not observe values with predictive potential in relation to pCR.

## Figures and Tables

**Table 1 ijms-26-02541-t001:** Association between clinical and biomarker variables and pathological complete response (pCR) in breast cancer patients undergoing neoadjuvant chemotherapy (NACT).

Variables	Number (n = 142)	pCR (n = 36)	Non-pCR (n = 106)	X^2^	*p*-Value
Age					
≤50	63	21	42	1.377	0.245
≥50	79	15	64		
BMI					
≤30 kg/m^2^	76	26	50	0.565	0.005
≥30 kg/m^2^	66	10	56		0.654
cT					
T1	74	14	60	0.756	0.065
T2	53	21	32		0.004
T3	15	1	14		0.076
cN					
N0	103	23	80	0.532	0.423
N1	39	13	26		
Grading					
G1	45	14	31	0.215	0.005
G2	71	17	54		0.004
G3	26	5	21		0.231
Histologic type					
Ductal	119	31	88	1.547	0.005
Lobular	17	3	14		0.654
Others	6	2	4		0.214
NLR					
<1	41	3	38	1.345	0.001
1–3	67	32	35		
>3	34	1	33		
PLR					
<120.45	87	29	58	21.145	0.000
>120.45	55	7	48		
SII					
<403.4	93	31	62	24.561	0.001
>403.4	49	5	44		
LMR					
<12.34	84	28	56	11.259	0.002
>12.34	58	8	50		
Ki-67					
Sub 14%	78	27	51	1.897	0.004
Peste 14%	64	9	55		

**Table 2 ijms-26-02541-t002:** Correlations between SII cut-off value and clinicopathological and molecular aspects of the cohort.

Variables	Number	SII < 403.4	SII > 403.4	X^2^	*p*-Value
Age					
≤50	63	26	37	5.235	0.001
≥50	79	31	48		
cT					
T1	74	29	45	0.154	0.654
T2	53	19	34		
T3	15	7	8		
cN					
N0	103	39	64	0.158	0.528
N+	39	11	28		
Ki-67					
Sub 14%	78	27	51	0.143	0.804
Peste 14%	64	18	46		
Molecular subtype					
Luminal A	68	33	35	0.198	0.001
Luminal B	39	17	22	0.211	0.004
HER2+	7	2	5	0.147	0.235
TNBC (six subtypes according to Lehmann classification)	28	16	12	0.174	0.325
basal-like 1 (BL1)	11	3	8	0.165	0.246
basal-like 2 (BL2)	6	2	4	0.689	0.758
immunomodulatory (IM)	4	3	1	0.256	0.785
luminal androgen receptor (LAR)	2	2	0	0.321	0.458
mesenchymal stem-like (MSL)	2	1	1	0.124	0.265
and mesenchymal (M)	3	1	2	0.239	0.324

**Table 3 ijms-26-02541-t003:** Evolutive dynamics of pre-therapeutic and post-therapeutic NLR, PLR, SII, LMR, and PNI in relation to pCR.

Variables	pCR	X ± S	T-Value	*p*-Value
Pre-NLR	yes	1.254 ± 0.24	−6.214	<0.001
	no	2.986 ± 1.24		
Pre-PLR	yes	114.23 ± 11.25	−5.214	0.002
	no	175.23 ± 42.23		
Pre-SII	yes	408.11 ± 98.23	−5.789	0.001
	no	821.24 ± 409.23		
Pre-PNI	yes	66.23 ± 6.23	0.741	0.624
	no	65.78 ± 2.31		
Pre-LMR	yes	24.14 ± 5.24	−3.145	0.001
	no	27.56 ± 2.56		
Post-NLR	yes	2.56 ± 1.35	−1.174	0.325
	no	2.61 ± 2.45		
Post-PLR	yes	151.12 ± 13.14	−1.149	0.127
	no	211.14 ± 24.14		
Post-SII	yes	491.24 ± 112.4	−1.611	0.178
	no	589.23 ± 231.56		
Post-PNI	yes	53.12 ± 3.56	2.147	0.231
	no	51.24 ± 45		
Post-LMR	yes	25.12 ± 2.34	−2.789	0.871
	no	38.25 ± 5.64		

PNI—objective index of inflammatory and nutrition status derived from serum albumin and lymphocyte counts.

**Table 4 ijms-26-02541-t004:** Univariate analysis of the association between biomarker ratios and pCR.

Variables	N (%) n = 142	pCR (%) n = 36	OR	95% CI	*p*-Value
PLR					
High	87	29	1.000	1.302–1.976	0.006 *
Low	55	7	1.467		
NLR					
High	49	17	0.987	0.871–1.785	0.004 *
Low	93	19	1.563		
LMR					
High	78	21	1.024	0.768–1.980	0.005 *
Low	64	15	1.893		
NLR/PLR					
High/High	39	19	1.000		0.001
Low/High	61	12	2.126		0.233
High/Low	9	3	No	No	No
Low/Low	33	2	11.456		0.02 *
Surgery					
RMM (radical modified mastectomy)-Madden procedure	45	3	1.000	0.211–7.276	0.698
BCS	72	22	1.177		0.004
OBCS	25	11	1.744		0.001

*-statistically significant.

**Table 5 ijms-26-02541-t005:** Association of patient/tumor characteristics to pCR in multivariate analysis.

Variables	OR	95% CI	*p*-Value
TN/HER2+ vs. Luminal	1.870	c	0.005 *
Grading G3 vs. G1/G2	0.551	0.004–23.508	0.026 *
Ki-67 ≥ 14% vs. Ki-67 ≤ 14	16.167	1.156–7.341	0.018 *
NLR low/PLR low vs. NLR high and/or PLR high	0.156	0.078–0.912	0.007 *
PLR low/PLR high	2.115	1.982–6.567	0.002 *
HR negative vs. HR positive	0.887	0.056–16.786	0.956
HER2 positive vs. HER2 negative	0.402	0.040–3.045	0.304
Chemotherapy + transtuzumab vs. Chemotherapy only	1.809	1.030–11.787	0.005 *

*-statistically significant.

## Data Availability

The original contributions presented in this study are included in the article. Further inquiries can be directed to the corresponding author.

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
