# Peer review of "Prognostic Significance of Peripheral Blood Parameters as Predictor of Neoadjuvant Chemotherapy Response in Breast Cancer"

_ijms, 2025, doi:10.3390/ijms26062541_

Round 1

Reviewer 1 Report

Comments and Suggestions for Authors

The study by Flaviu et al. analyzed 142 breast cancer patients from Romania, focusing on their clinical and pathological characteristics and various blood parameters before treatment.  Statistical analyses were performed to determine the significance of associations between these parameters and the likelihood of achieving pCR. This study has much clinical value, and the efforts behind it are evident. The study is supported by a robust statistical analysis including ROC curves. A significant implication is that utilizing peripheral blood parameters can enhance the ability to predict treatment outcomes and tailor neoadjuvant chemotherapy effectively in breast cancer. Despite this manuscript's original contribution, this reviewer recommends a major revision. The manuscript is challenging to read and follow. These are some points the authors might want to consider:

1.      The abstract contains at least 6 acronyms, each representing a variable. It is challenging to follow them.

2.      NAC and NACT are used interchangeable. NAC is also an acronym for N-acetylcysteine, so NACT is a preferred acronym.

3.      Abstract the methods, first sentence, standardized for what?

4.      Interrelational analysis needs to be defined.

5.      The abstract needs to present the central question or hypothesis. This is implied in the last sentence of the Conclusions in the abstract.

6.      A P-value of 0.000 is impossible in statistics, redefine, e.g. p<0.001.

7.      PNI not defined.

8.      Why is HER-2 positive an unfavorable molecular type? It should be favorable because of the availability of Herceptin and other drugs.

9.      The legend of Table 1 needs to be expanded to be self-explanatory.

10. Table 2: It is recommended that all 6 TNBC subtypes be grouped into one TNBC and that the statistical analysis be done based on one TNBC and the other three molecular subtypes.

11. Table 4-Define RMM-Madden procedure.

12. It is strongly recommended that the Discussion be divided into various topics using subheadings, as this will improve readability.

13. Methods- How was the molecular stratification of the BC patients done?

14. How were hormone receptor markers determined? Based on what methods?

15. What volume of blood was extracted from patients?

16. Clarify that the hospital is in Romania in the methods section.

Comments on the Quality of English Language

The quality of English is satisfactory.

Reviewer 2 Report

Comments and Suggestions for Authors

Summary:
In this paper, the authors perform statistical analysis to study the correlations between lots of clinical characteristics (including peripheral blood biomarkers) and pathological complete response (pCR) from 142 breast cancer patients, who have received neoadjuvant chemotherapy. The results demonstrates that some blood parameters, such as platelet-to-lymphocyte ratio (PLR), neutrophil-to-lymphocyte ratio (NLR), lymphocyte-to-monocyte ratio (LMR), and systemic immune-inflammatory index (SII), are highly correlated with pCR, indicating they can be used as predictors of pCR.

Major:
+ The methodology is well designed and described.
+ The authors scientifically performed analysis on several clinical characteristics, and clearly presented the results.
+ The findings may have potential applications in clinical practice.
- The discussion section in this paper basically introduces some related works, rather than providing a detailed analysis of the study’s own results.
- The authors reused sentences from the main text in the abstract, which made the abstract lack logical flow and hard to read.

Minor:
- The terms, pathological complete remission (pCR) and pathological complete response (pCR), share the same abbreviation; however, the authors did not clearly differentiate between them, which may lead to ambiguity in interpretation.
- There are grammatical errors in the paper. For example, in abstract, "The analytical parameters were systematically and standardized".
- Typo in keywords: Pathological complete respons -> response

Reviewer 3 Report

Comments and Suggestions for Authors

I consider that the paper “Prognostic Significance of Peripheral Blood Parameters as Predictor of Neoadjuvant Chemotherapy Response in Breast Cancer” is interesting, since it is very important to find biomarkers that predict response to neoadjuvant chemotherapy, however I suggest improving the following aspects:

In the abstract it would be convenient to add that pCR stands for pathological complete remission, since it is not a frequently used abbreviation.

On line 73 it is not necessary to add that pCR stands for pathological complete remission since it was made explicit on line 65 and in the keywords.

Unify the abbreviation to be used for the term “neoadjuvant chemotherapy” since in some lines it is mentioned as NAC therapy and in others as NACT. Both are used in the bibliography.

It seems interesting that the Pre-therapeutic values of NLR, PLR, SII and LMR are lower in the qCR group, appearing to be good markers of response prediction, but these differences are lost in Post-treatment. What could be the reason for this finding and what could be the clinical implication?

In the discussion I consider that although they review the available literature on the field, there is no deeper reflection on the contributions of their findings.

I consider that it is necessary to emphasize what is innovative about the work and what are the contributions with respect to others of similar characteristics mentioned in the discussion.

It would also be interesting to discuss how this could be transferred to the clinic and what are the future perspectives and lines of research that could be opened. Finally... If the values predict that patients will not present qCR, what could be the behavioral implications of these findings?

Round 2

Reviewer 1 Report

Comments and Suggestions for Authors

The revised version of the manuscript appears to have addressed several concerns/suggestions made by this reviewer. No letter explaining the changes was received. However, the most important scientific suggestion requested for the revised manuscript was not done. "Table 2: It is recommended that all 6 TNBC subtypes be grouped into one TNBC and that the statistical analysis be done based on one TNBC and the other three molecular subtypes". This issues is still pending.

Reviewer 2 Report

Comments and Suggestions for Authors

All my concerns were addressed in the revised version.

Round 3

Reviewer 1 Report

Comments and Suggestions for Authors

The authors have satisfactorily addressed most of the concerns/questions of this reviewer.